# Potentials and limitations in the application of Convolutional Neural Networks for mosquito species identification using wing images

Kristopher Nolte[1]*, Jan Baumbach[2], Christian Lins[3], Jens Johann Georg Lohmann[2], Philip Kollmannsberger[4], Felix Gregor Sauer[1‡], Renke Lühken[1‡]

**1** Arbovirus and Entomology Department, Bernhard Nocht Institute for Tropical Medicine, Hamburg, Germany, **2** University of Hamburg, Institute for Computational Systems Biology, Hamburg, Germany, **3** Department of Computer Science, Hamburg University of Applied Sciences, Hamburg, Germany, **4** Physics Department, Heinrich Heine University Düsseldorf, Düsseldorf, Germany

‡ Shared last author.
* kristopher.nolte@bnitm.de

## Abstract

This study addresses the pressing global health burden of mosquito-borne diseases by investigating the application of Convolutional Neural Networks (CNNs) for mosquito species identification using wing images. Conventional identification methods are hampered by the need for significant expertise and resources, while CNNs offer a promising alternative. Our research aimed to develop a reliable and applicable classification system that can be used under real-world conditions, with a focus on improving model adaptability to unencountered devices, mitigating dataset biases, and ensuring usability across different users without standardized protocols. We utilized a large, diverse dataset of mosquito wing images of 21 taxa and three image-capturing devices (N = 14,888) and a preprocessing pipeline to standardize images and remove undesirable image features. The developed CNN models demonstrated high performance, with an average balanced accuracy of 98.3% and a macro F1-score of 97.6%, effectively distinguishing between the 21 mosquito taxa, including morphologically similar pairs. The preprocessing pipeline improved the model's robustness, reducing performance drops on unfamiliar devices effectively. However, the study also highlights the persistence of inherent dataset biases, which the preprocessing steps could only partially mitigate. The classification system's practical usability was demonstrated through a feasibility study, showing high inter-rater reliability. The results underscore the potential of the proposed workflow to enhance vector surveillance, especially in resource-constrained settings, and suggest its applicability to other winged insect species. The classification system developed in this study is available for public use, providing a valuable tool for vector surveillance and research, supporting efforts to mitigate the spread of mosquito-borne diseases.

**Data availability statement:** Images used in this study were previously published on a data descriptor and are available under CC BY 4.0 license at https://doi.org/10.6019/S-BIAD1478. Scripts which were used to train models are available on the project git page: https://github.com/KNolte19/MosquitoWingClassifier_publication. Downloadable and installable docker application can be accessed on the applications' git page: https://github.com/KNolte19/MosquitoWingClassifier. A demonstrator of the application is hosted on https://balrog.bnitm.de/, Instructions on how to access the website can be found on the applications repository: https://github.com/KNolte19/MosquitoWingClassifier.

**Funding:** This project is funded through the Federal Ministry of Education and Research of Germany (01Kl2022 to KN, FGS, RL). The funders had no role in study design, data collection and analysis, decision to publish, or preparation of the manuscript.

**Competing interests:** The authors have declared that no competing interests exist.

## Author summary

Our study investigates the application of Convolutional Neural Networks (CNNs, refer to Table A in S1 File for glossary) for mosquito species identification using wing images, providing a scalable alternative to traditional, expert-dependent methods. By developing a robust preprocessing pipeline, we standardized images from various capturing devices to enhance the model's adaptability and partially address inherent dataset biases. Our CNN-based approach was trained on a diverse dataset representing 21 mosquito taxa and achieved an impressive overall accuracy of 98.3%, effectively distinguishing between even morphologically similar species. This work demonstrates significant potential for real-world vector surveillance, especially in resource-limited settings where rapid, accurate identification is critical for public health. The methodology not only streamlines species identification but also opens avenues for its application to other winged insect species. Ultimately, we offer a functional, openly accessible tool designed to support researchers and public health officials in monitoring and mitigating the spread of mosquito-borne diseases.

## 1 Introduction

The global burden of mosquito-borne pathogens is of pressing concern. Diseases such as malaria, dengue fever, or chikungunya cause significant morbidity and mortality worldwide, straining the public and veterinary health systems and impeding socio-economic development in affected areas [1]. Furthermore, climate change and increased global trade contribute to the spread of mosquitoes and their associated pathogens [2].

The vector capacity of a mosquito species is a function of its ecology, behaviour, and vector competence, making accurate species identification a prerequisite for effective vector surveillance, research and control measurements [3]. Conventional identification techniques, such as morphological analysis, are time-consuming and require specialist expertise, which is less and less available [4]. Alternative methods like molecular assays, can be prohibitively expensive and challenging to use in low-cost settings [5]. This concurs with the statement of the European Centre for Disease Prevention and Control, which underscores that a principal hurdle to the establishment of robust vector surveillance systems is the scarcity of trained experts and financial resources available [6].

Within the context of insect species identification, machine learning (ML) driven methods have shown considerable promise, offering advancements that could automatize the identification process. Especially CNNs have emerged as a powerful tool for identifying insects based on images, e.g., tsetse flies [7], sandflies [8] or bees [9]. For mosquitoes, several CNN models are presented in literature which are capable of classifying mosquito species based on images with varying scope and performance. For instance, Goodwin et al. (2021) achieved a remarkable 97% accuracy

in identifying 17 mosquito species using standardized images of their bodies taken with a specifically designed imaging tower and Zhao et al. (2022) were even capable of distinguishing between species of the *Culex pipiens* complex, which cannot be morphologically differentiated by human eye [10]. Although using images of whole mosquitoes may appear intuitive, leveraging wings for species classification provides distinct advantages. In our previous research, we demonstrated that using wing instead of body images not only improves classification performance but also reduces data demands for model training [11]. Wings are nearly two-dimensional, which makes the imaging process easily standardizable, as it does not require images from multiple angles. Moreover, wings and their vein patterns remain unaffected by the mosquito's physiological status, such as being blood-fed or gravid [12].

Traditional ML pipelines, while effective in many scenarios, frequently fall short in real-world deployment [13–15]. ML development typically follows a standardized framework involving model specification, a training dataset, and an evaluation procedure that assumes identical distributions of training and deployment [15]. However, this approach does not consider the potential for inherent biases and underspecifications within trained models. For instance, shortcut learning, as described by Geirhos et al. (2020), refers to a phenomenon where ML models learn to rely on simple, exploitable spurious correlations in the data ("shortcuts") rather than the intended, more complex representations. For example, a model designed to identify cows in images might achieve high performance by leveraging the background context, such as associating the presence of grass with cows [14]. These shortcuts often result in models that perform well on specific datasets but fail to generalize to new, unseen data [15,16]. Another aspect of underspecifications is that CNN models are typically not specified to handle novel devices with varying imaging characteristics [15], a limitation also evident in studies on insects [17,18], which we have also observed in our previous study on mosquito species identification [11]. Moreover, particularly in the context of image classification, deep learning methods are very data demanding. Thus, while the concept of a homogeneous dataset may seem ideal for a proof-of-concept study or in-house solutions, its practical implementation beyond such controlled environments is often unfeasible. These discrepancies can lead to suboptimal performance in real-world conditions, and can thereby explain the often observed gap between proof-of-concept studies and practical applications in vector surveillance [11].

Thus, there is a need for robust, open-access systems that can reliably identify mosquitoes under various image collection settings. Such an application empowers researchers and public health authorities to quickly and accurately classify mosquito species, reducing reliance on labour-intensive morphological or molecular methods. Furthermore, it could enable a broader spectrum of applicants, including non-entomologists to identify mosquitoes. Building upon our previous proof-of-concept studies using CNNs for mosquito species identification based on wing images [11, 19], this work advances towards practical, real-world deployment. Earlier research demonstrated the feasibility of species classification using models such as EfficientNet on curated datasets [11,19]. However, those studies did not fully address the challenges encountered in field applications, such as variability in image acquisition, heterogenous training sets, and usability by non-specialist users. In this study, we build upon our previous work by addressing three practical challenges that limit real-world applicability: (1) ensuring robust model performance across images from previously unseen devices, (2) reducing biases stemming from heterogeneous and variably sourced training data, and (3) improving usability for non-expert users in uncontrolled environments. To this end, we developed an open-access, scalable classification system trained on large, diverse dataset comprising mosquito wing images from multiple mosquito taxa and imaging devices. A key component of our approach is an upstream preprocessing pipeline that automatically standardizes images and removes irrelevant visual information. We evaluate the system via four complementary approaches: in-distribution testing, simulation-based analysis of preprocessing effectiveness, interpretability through explainable AI techniques (Grad-Cam), and a practical feasibility study. Ultimately, this study provides a robust, deployable tool for mosquito identification designed to support both researchers and public health practitioners in the field. The final application is published in a GitHub repository (https://github.com/KNolte19/MosquitoWingClassifier) and is hosted as web-service on https://balrog.bnitm.de. All code to reproduce the results and figures in this manuscript can be found in second repository (https://github.com/KNolte19/MosquitoWingClassifier_publication).

## 2 Materials and methods

### 2.1 Ethics statement

Human participants were involved solely to capture images of mosquitoes for the feasibility study. Based on the Hamburg University of Applied Sciences Ethics Application self-assessment checklist, the study qualified as exempt from formal ethics committee review under applicable regulations.

### 2.2 Dataset construction

We used a total of 14,888 images of female mosquito wings from an open-access dataset [20] we previously published, which includes 8,947 individual specimens across nine genera: *Aedes* (*Ae.*), *Anopheles* (*An.*), *Culex* (*Cx.*), *Culiseta* (*Cs.*), *Coquillettidia* (*Cq.*), *Armigeres* (*Ar.*), *Mansonia* (*Ms.*), *Uranotaenia* (*Ur.*), and *Toxorhynchites* (*Tx.*). These specimens were categorized into 72 taxonomic units, including single species, species pairs, groups, and complexes as defined in the original dataset. For 48.2% of the specimens, both wings were imaged, and 11.2% of all wings were photographed using two different devices.

To prepare the data for CNN-based classification, each mosquito specimen was assigned a single label corresponding to a defined taxonomic unit. Mosquito taxonomy presents inherent challenges as certain species cannot be reliably distinguished based on morphology or even molecular barcoding, especially among females [4]. To account for these limitations and to ensure label reliability, we adopted a taxonomy-informed labelling strategy introduced in our prior work [20]. Specifically, species that are morphologically or genetically indistinguishable were merged into broader taxonomic units. Namely *Ae. annulipes* group, *Ae. cinereus–geminus* pair, *Ae. communis–punctor* pair, *An. claviger–petragnani* pair, *An. maculipennis* complex, *Cs. morsitans–fumipennis* pair, *Cx. torrentium–pipiens* complex, and *Cx. vishnui* group. These aggregated taxa are treated as single, coherent labels. The Integrated Taxonomic Information System identifiers for each taxa label are provided in Table C in S1 File).

To ensure sufficient data for robust model training, we excluded underrepresented taxa by grouping those with fewer than 80 images under a single "other" label (N = 34 taxa). This threshold was chosen based on prior experiments [11], which demonstrated poor model performance on classes with very limited sample sizes. After aggregation and filtering, the final dataset comprised 21 distinct classification labels (see Fig 1). Images were captured using three different setups: two stereomicroscopes (Olympus SZ61 with DP23 camera (63.5%) and Leica M205 C (25.7%)) and a smartphone-mounted macro lens (Apexel-24XMH on an iPhone SE, 3rd Generation; 10.8%).

We employed a six-fold cross-validation strategy to assess the performance and transferability of our CNN models (Fig 2). The dataset was partitioned into six approximately equal folds (~2,500 images per fold), with stratification to maintain a balanced distribution of labels. Crucially, the splitting was performed at the specimen level rather than the image level to prevent data leakage, as some specimens were associated to multiple images (e.g., both wings or repeated image captures with different devices). The cross-validation process was divided into two main phases. 1) Hyperparameter tuning: We reserved the first fold exclusively for model tuning. This fold served as a validation set, while the remaining five folds were used for training during the tuning phase. 2) Model evaluation: Once the optimal hyperparameters were selected, we conducted five independent training runs. In each run, one of the remaining five folds served as the test set, while the other four folds were used for training. This two-stage strategy allowed us a more reliable estimate of model accuracy across multiple data compositions. Comprehensive details about the images, including their labels, capture devices, and sources, are provided in the publications' GitHub repository (https://github.com/KNolte19/MosquitoWingClassifier_publication) with a complete table describing all data (GitHub Repository/database_reference.xlsx).

### 2.3 Preprocessing pipeline and augmentation

We developed an image preprocessing pipeline, aiming to reduce inherent biases in our heterogenous dataset and improve the model's applicability to images captured by novel devices (Fig 3). The pipeline aims to eliminate undesired

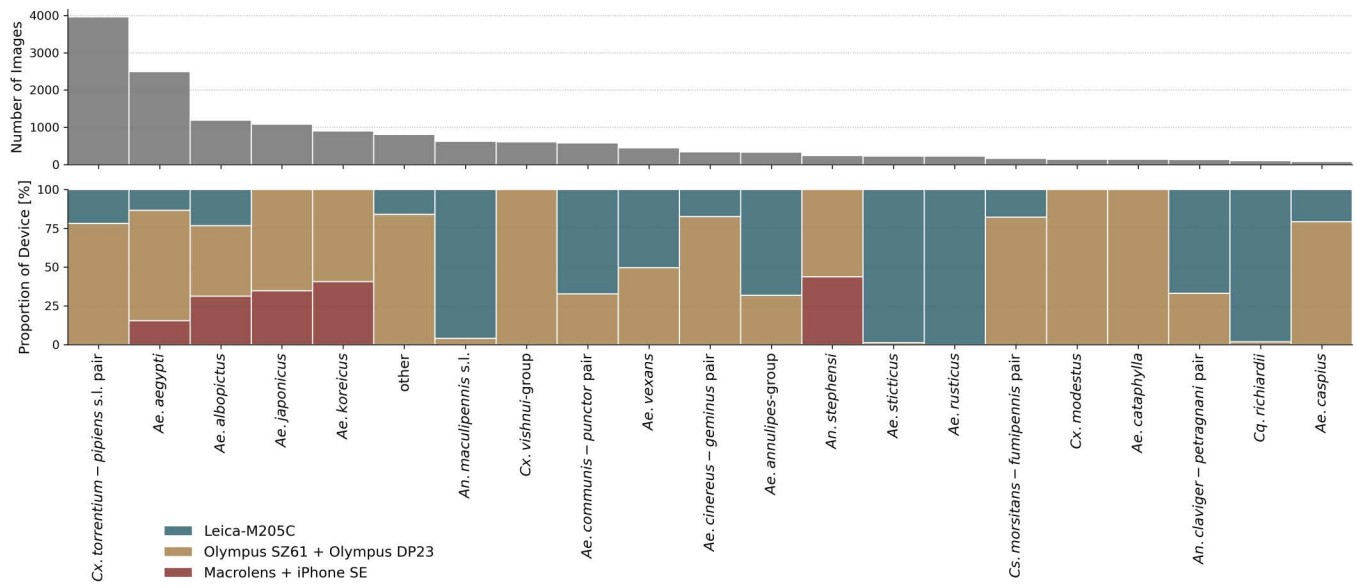

**Fig 1. Dataset distribution Number of images per taxa label (top) and proportion of image capture devices per label (bottom).**

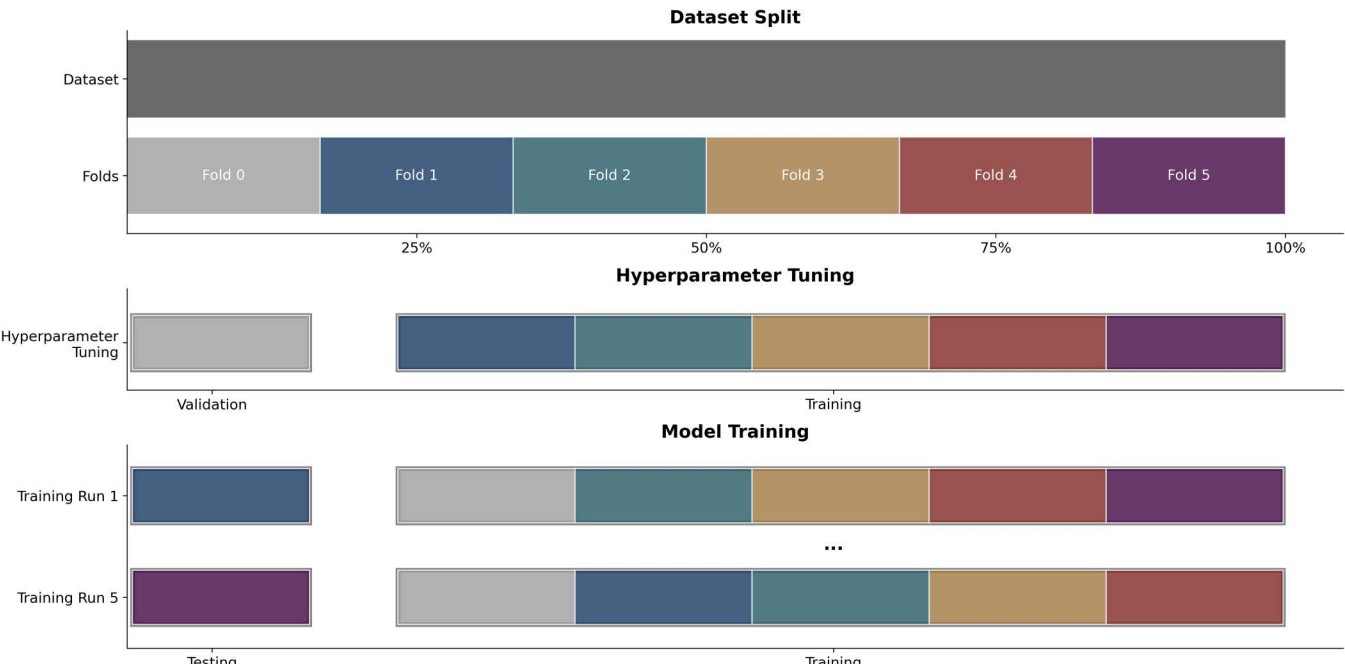

**Fig 2. Plot illustrating the division of the image dataset of hyperparameter tuning and model training.** Data was separated into six folds (Top). First, the model hyperparameters were optimized by utilizing the first fold (fold 0) as the validation set (Middle). The bottom sections depict the training strategy, showcasing cross-testing across folds to generate five unique dataset compositions, with one fold designated as a testing set in each training run.

PLOS Computational Biology

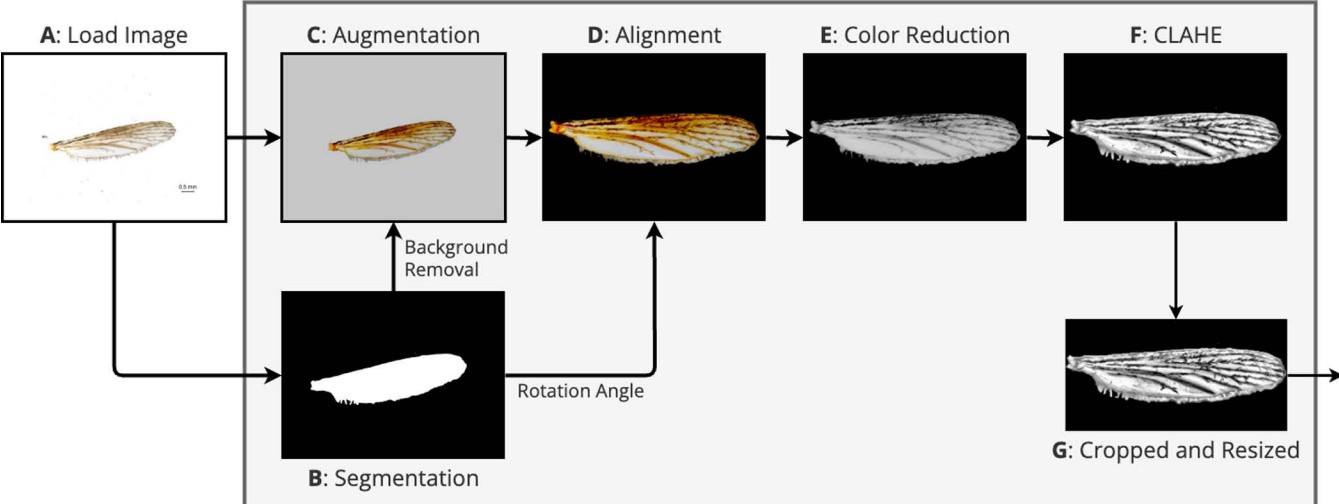

**Fig 3. Schematic representation of the image preprocessing pipeline.** The process starts with the loading of the original wing image (A). Background removal is performed to isolate the wing using a segmentation mask (B). The image is augmented by an augmentation pipeline during training and in deployment (C). The augmented wing is then rotated to a standardized horizontal orientation based on the calculated rotation angle (D). The image is converted to greyscale to reduce colour variability (E). CLAHE is applied to enhance contrast and highlight important features (F). Finally, the processed image is resized to a uniform dimension, making it suitable for input into the CNN model (G).

features such as varying lighting conditions, background, and colour variations. In the first step, the images are loaded (Fig 3, step: A) and hereafter a U-Net model from the Rembg Python library is employed in the second step to create a segmentation mask of the wing and background to remove the latter (step: B, github.com/danielgatis/rembg). In the third step, the image is augmented (step: C). To implement this, we utilized augmentation functions from the Albumentations library to construct an augmentation pipeline consisting of 13 layers: ISONoise, PlanckianJitter, ImageCompression, Defocus, RandomGamma, MotionBlur, Downscale, ColorJitter, ChannelDropout, MultiplicativeNoise, RandomZoomOut, RandomHorizontalFlip, and RandomRotation [21]. In the fourth step, orientation of the wings is horizontally aligned by using region properties measured from the segmentation mask of the wing (step: D), implemented with sci-kit-image [22]. This alignment adjusts the horizontal orientation of wings across different images, reducing variability and aiding consistent feature extraction. To reduce the impact of colour variations, which are generally not very informative for traditional mosquito species identification by wings [19], the images are converted to greyscale (step E). This conversion simplifies the image while retaining essential structural information, improving the robustness of the model to varying colour changes due to different imaging settings. Contrast Limited Adaptive Histogram Equalisation (CLAHE) with a high clip limit (0.5) is applied (step F) before median filtering [23]. CLAHE enhances image contrast and improves the visibility of the wing vein pattern. Median filtering smooths out noise and further enhances the clarity of the wing details. Lastly, images are cropped and resized to a resolution of Y:192 and X:384 to ensure they are suitable for input into the CNN model (step G). This size is chosen because it keeps enough detail for the model to recognize important features while also reducing the computational effort required for training.

Preprocessing steps, such as image normalization and resizing, were applied consistently across all stages: training, testing, and deployment. In contrast, data augmentation was applied only during training to increase dataset variability and improve model robustness. For deployment, we used test-time augmentation, where multiple augmented versions of the input image were evaluated and their predictions averaged. During all model evaluations, no augmentation was applied.

## 2.4 Classifier training

A training setup was developed, encompassing the pipeline and training procedure using the Python programming language and the PyTorch library. The Python scripts and a list of all used libraries and their versions can be found on the publications GitHub repository (GitHub Repository).

The issue of imbalanced data distribution is addressed by resampling a uniform distribution from the training dataset in each training epoch. The training process was facilitated by using pre-trained CNNs and the fine-tuning learning strategy. The EfficientNetB0 model architecture without the original classification head, pre-trained on the *ImageNet* dataset, was utilized as a feature extractor [24,25]. EfficientNet was selected as it showed a high performance in prior studies on mosquito wing classification, as well as its favourable trade-off between accuracy, training speed, and parameter efficiency [26]. Additional layers, namely a Dropout and a Dense Layer, were appended for classification.

Hyperparameters such as learning rate, batch size, and image size were systematically optimized through experimental iterations and evaluation of the validation set. For model training, categorical cross-entropy emerged as a loss function for the classification task. The Adam optimizer with weight decay was employed to optimize the learned parameters during model training [27]. A complete list of hyperparameters resulting in the final models can be found in Table B in S1 File. After the hyperparameters were determined, five models were trained and tested on five folds to acknowledge the stochastic nature of ML models (Fig 2). The mean balanced accuracy and macro F1-score of the five models are reported with a 95% confidence interval (CI95%) on the testing set (refer to Table G in S1 File for definition of metrics).

## 2.5 Robustness experiments

To assess the robustness of our mosquito classification system under real-world deployment scenarios, we conducted two experiments targeting key sources of variation: unfamiliar image-capturing devices and dataset-induced bias. These experiments were designed to evaluate the effectiveness of our image preprocessing pipeline in enhancing generalization and reducing reliance on device-specific artifacts.

We used a systematically collected dataset of 3,027 wing images from 793 female specimens of four Aedes species (*Ae. aegypti*, *Ae. albopictus*, *Ae. koreicus*, and *Ae. japonicus*). This dataset was previously published [11] and is also a subset of the larger dataset described in section 3.2. Each specimen in this subset was imaged using two distinct devices: a stereomicroscope (Olympus SZ61 with a DP23 camera) and a smartphone with a macro lens (Apexel-24XMH on an iPhone SE). As such, this dataset enabled controlled evaluation of cross-device generalization and allowed the intentional introduction of imaging biases in a structured manner.

To test the impact of preprocessing on cross-device generalization, we generated four versions of this dataset, each corresponding to an increasing level of preprocessing applied to the images: (step A referring to Fig 3) original images resized and padded; (C) image versions A with data augmentation; (D) background removal, horizontal wing alignment, augmentation and resizing without padding; and (G) the full preprocessing pipeline including color reduction and CLAHE. These datasets were split into five folds following the same specimen-level stratified protocol described in Section 3.2 (see Fig 2), except that no fold was reserved for validation, as hyperparameters remained fixed. Yet, due to the smaller dataset size, we reduced training epochs to 12 and froze the first half of the model layers to mitigate overfitting. Five models were trained for each level of preprocessing.

In the first experiment—the novel-device experiment—we investigated the model's ability to generalize to images from devices not seen during training. We constructed two separate dataset compositions, one containing only microscope images and the other only smartphone images. For each composition, we trained five models using the five-fold cross-validation scheme, training on four folds and testing on the held-out fold. Each trained model was then additionally evaluated on the entire alternative composition captured with the other device. This setup allowed us to assess the effect of image-capturing device differences on model performance and evaluate whether preprocessing mitigated the loss in accuracy when faced with previously unseen imaging characteristics.

In the second experiment—the bias experiment—we examined whether our preprocessing pipeline could reduce model reliance on spurious correlations between image characteristics and class labels. We constructed an intentionally biased dataset composition in which *Ae. aegypti* and *Ae. koreicus* were represented only by microscope images, while *Ae. albopictus* and *Ae. japonicus* were represented only by smartphone images. This introduced a confounded association between species and device type. Using the same five-fold cross-validation strategy, we trained five models on this biased dataset composition. Each model was tested both on its held-out fold from the biased dataset composition and on the other images with the inverse species–device combinations (e.g., *Ae. aegypti* imaged with a smartphone instead of microscope). This additional test set enabled us to quantify the extent to which the model relied on device-related features when making predictions, and to what degree preprocessing reduced this bias. Full details of the dataset structure and device distributions are available in Tables D-F in S1 File.

## 2.6 Model exploration

We employed Gradient-based Class Activation Mapping (GradCam) to gain insights into the decision-making process of the CNN models. GradCam calculates the gradients of the target class for the final convolutional layer of the CNN, highlighting regions with high gradients to produce a heatmap visualization of the most discriminative image regions [28]. Recognizing the potential challenges in interpreting GradCam heatmaps [29], we applied GradCam to multiple images in the test dataset, averaging the resulting activation maps and images. This approach was facilitated by our preprocessing pipeline, which ensured that all images were consistently aligned. For this analysis, we utilized the best-performing model and mosquito wing images from the Sauer et al. (2024) study, as this dataset provided uniformly oriented images of the right wing. We focused on six taxa labels from the dataset as these provided the most images: *Ae. cinereus-geminus*-pair, *Ae. communis-punctor*-pair, *Cq. richiardii*, *Ae. rusticus*, *Ae. sticticus*, and *Ae. vexans*. For each taxon, we randomly selected 12 images from the testing fold, applied GradCam to these images, and subsequently averaged both, the GradCam activation maps and the original images, to create a composite visualization of the most discriminative regions for each taxon.

To further investigate the decision-making process of the CNN models, we applied Uniform Manifold Approximation and Projection (UMAP) to the feature maps generated by the best-performing CNN model. UMAP is a dimensionality reduction technique that enables visualization of high-dimensional data in a lower-dimensional space while preserving the underlying structure and relationships [30]. By applying UMAP to the feature maps, we aimed to uncover patterns and relationships within the data that may not be apparent in the original high-dimensional space.

## 2.7 Feasibility study

A feasibility study was conducted to assess the applicability of the CNN model under real-world conditions. Four participants were tasked to independently capture images of mosquito wings of 10 taxa labels, comprising a total of 78 wings: *Ae. aegypti* (8 images), *Ae. albopictus* [8], *Ae. japonicus* [8], *Ae. koreicus* [8], *Ae. vexans* [8], *Cx. pipiens* s.l./*Cx. torrentium* [8], *Cx. tritaeniorhynchus* [8], *An. maculipennis* s.l. [6], *An. stephensi* [8], *Cq. richiardii* [8]. Additionally, we included 8 wings belonging to the "other" label (*An. coustani*-group). Participants utilized a stereomicroscope (Olympus SZ61, Olympus, Tokyo, Japan) equipped with a camera (Olympus DP23, Olympus, Tokyo, Japan). Subsequently, all 78 wings from the feasibility study were again photographed by one single observer using an imaging device which was neither represented in the training nor the testing dataset (Zeiss Stemi 2000-C stereomicroscope, Oberkochen, Germany, equipped with a Nikon Coolpix P950, Tokyo, Japan). Finally, all captured images underwent processing through the image preprocessing pipeline. The processed images were then classified with all 5 models.

## 2.8 Model deployment

To make the image preprocessing pipeline and the model accessible, we developed a user-friendly application using the Python framework Flask [31]. This application allows users to upload images of mosquito wings, which are then

processed through our imaging pipeline. Each uploaded image is augmented multiple times, a process known as test-time augmentation, before being analyzed by the best-performing CNN model. Test-time augmentation enhances the model's robustness by generating several modified versions of each input image [32]. Following the augmentation, predictions from four different augmented versions are averaged. Averaging predictions over these augmented images reduces the impact of noise and variations in the original image, leading to improved accuracy and more stable performance. The resulting average is then calibrated using a logistic regression model to produce an output between 0 and 1, representing the model's uncertainty [33].

Once the images are processed, users receive a display of the predictions along with a CSV file containing the averaged and calibrated outputs. To ensure consistency across various deployment environments, we employed Docker to containerize the entire classification system. This encapsulation of all dependencies and configurations simplifies installation and guarantees seamless execution, regardless of the operating system or environment. The application, together with installation instructions, is available on GitHub (GitHub Repository).

## 3 Results

### 3.1 Classifier performance

Following the selection of the final hyperparameters, five models were trained on the five training configurations and subsequently evaluated on the remaining images of the respective testing fold (Table 1). The CNN models exhibited high performance, achieving an average balanced accuracy of 98.2% (CI95%: 97.8 - 98.6) on the testing sets (Fig 4). The lowest performing model (testing fold: 1) achieved 97.9% while the highest performing model (testing fold: 5) reached 98.6% balanced accuracy. The average macro F1-Score was 97.6% (CI95%: 96.9 - 98.4). The model that has been tested on fold 2 has shown the best performance in terms of balanced accuracy with 98.6%. In total, summarizing all testing folds, 214 of 12323 images (1.7%) were misclassified. Among the misclassified images, 26 (12%) underwent undesirable preprocessing, while 30 wings (14%) exhibited damaged features. The classification accuracy varied only a little across different taxa, with 2 out of 21 labels achieving flawless classification (100%), and 19 labels attaining accuracy levels exceeding 95%. The CNN models demonstrated lower average accuracy (≤ 95%) only for the "other" taxon label.

### 3.2 Robustness experiments

In the novel-device experiment, we assessed the models' applicability to images captured with unfamiliar devices by testing the performance under different levels of preprocessing. The models' performance on images from a new device improved with higher levels of preprocessing, while their performance on the original training device remained stable (Fig 5A and 5B). The highest performance on a novel device is achieved by the full processing method. It resulted in an average accuracy of 90.0% (CI95%: 87.2 - 92.9) and 94.5% (CI95%: 92.2 - 96.7) when trained only on microscope or macro lens images, respectively. This means a performance drop of 7.8% for the microscope-trained models and 2.1% for

**Table 1. Composition of training and testing data in addition to performance results of each model.**

| Testing Fold | N images used for Testing | N images used for Training | % images used for Training | Accuracy [%] | Balanced Accuracy [%] | Macro F1 Score [%] |
|---|---|---|---|---|---|---|
| 1 | 2469 | 12419 | 83.4 | 97.9 | 97.9 | 96.7 |
| 2 | 2486 | 12402 | 83.3 | 98.7 | 98.6 | 98.2 |
| 3 | 2439 | 12449 | 83.6 | 98.3 | 98.4 | 98.1 |
| 4 | 2492 | 12396 | 83.3 | 98.1 | 97.9 | 97.5 |
| 5 | 2437 | 12451 | 83.6 | 98.4 | 98.1 | 97.6 |

PLOS Computational Biology

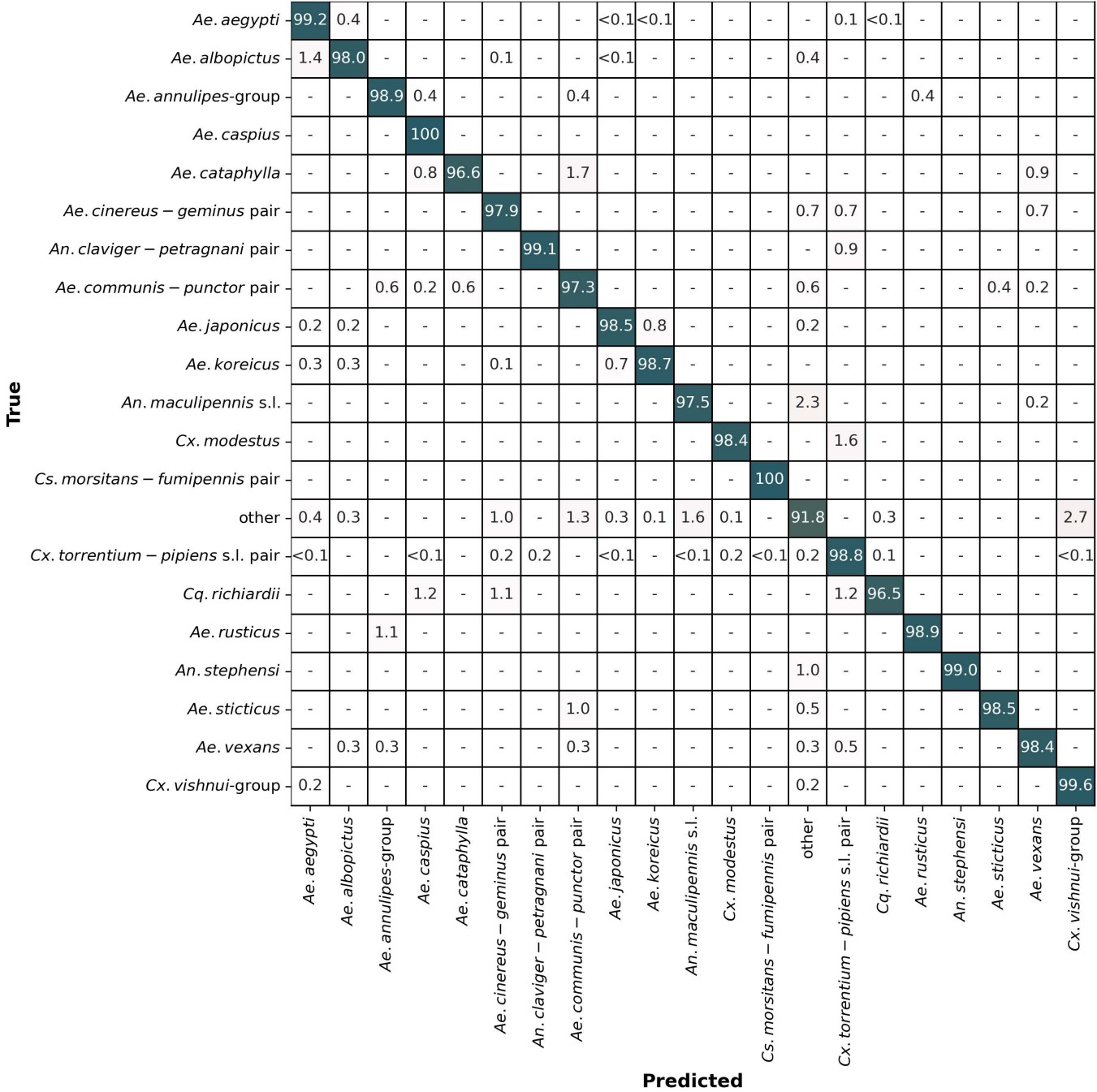

**Fig 4. Confusion matrix showing the average accuracy of the five CNN models.** Each cell represents the proportion of predictions made for each true taxon (%).

the macro-lens-trained models. In contrast, with minimal preprocessing (only resizing), the performance drop was much larger: 75.2% for the microscope-trained models and 62.4% for the macro-lens-trained models.

In the second experiment, we introduced bias into the training dataset by using exclusively smartphone images for two mosquito species and microscope images for the other two species. When tested on the same species device

combination as in the training dataset we observed a nearly perfect classification performance (>99%) on all preprocessing levels. Yet, as anticipated, the models performed poorly when tested on novel combinations of species and imaging devices. Without preprocessing, the models failed to correctly predict any images with a novel species-device combination (Fig 5C and 5D). However, increasing the level of preprocessing resulted in marginal improvements in model performance and a substantial boost was observed when full preprocessing was applied. For images captured with a smartphone, the models achieved an average accuracy of 35.7% (CI95%: 25.1 - 46.3) on novel species-device combinations when full preprocessing was applied (Fig 5C). A similar, but less pronounced, improvement was observed with microscope images, where accuracy increased by 18.7% (CI95%: 12.3 - 25.2) under full preprocessing (Fig 5D).

### 3.3  Model evaluation

GradCam analysis was performed to provide insights into the decision process of the CNN models for six species. When wing images were averaged, the resulting depictions clearly show individual veins, demonstrating the effectiveness of the preprocessing pipeline in consistently aligning the wings (Fig 6). The GradCam images revealed variations in the importance of regions used for classification across different species. Across all species, the visualizations consistently emphasize structural patterns near the central regions of the wing. The primary regions of importance appear to correspond to intricate vein arrangements and interveinal membrane structures. For all taxa examined, the highest activation regions are observed along the medial section of the wing, where vein branching is most complex, while peripheral areas (distal tips

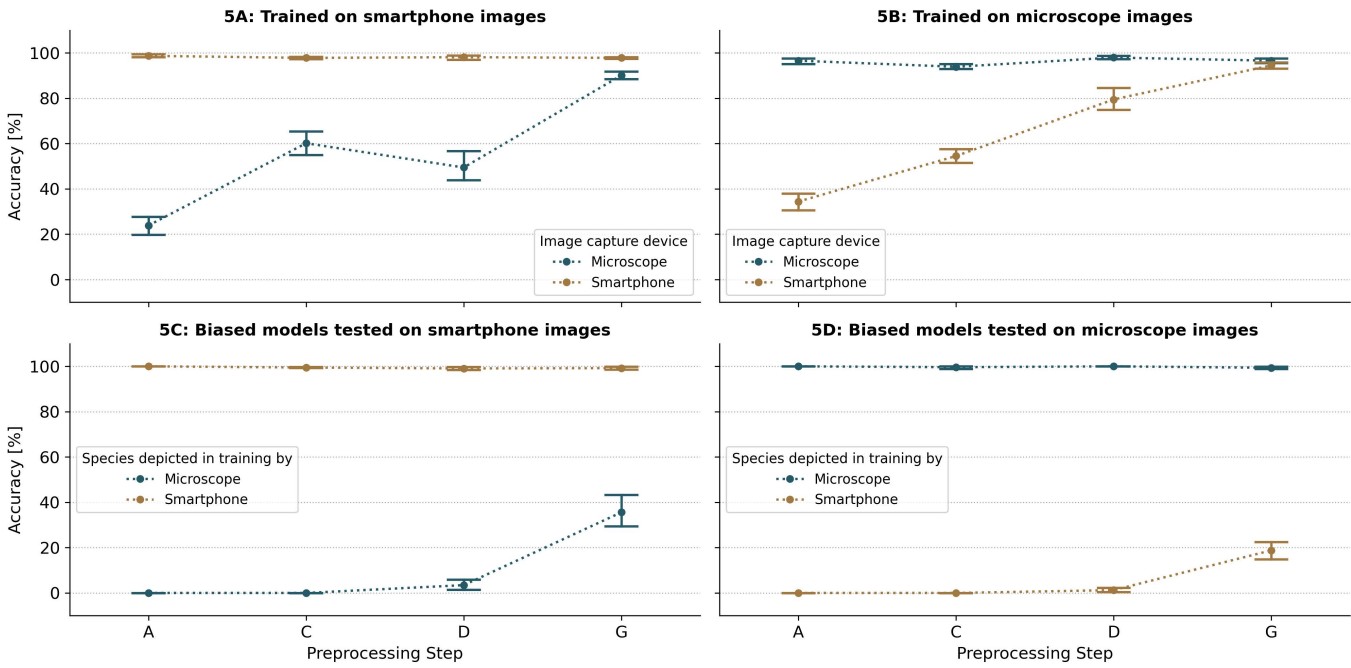

**Fig 5.  Impact of preprocessing on model robustness across device and bias conditions.** This Fig presents the accuracy (mean±95% confidence interval) of CNN models trained and tested under different conditions designed to evaluate model robustness. Four levels of image preprocessing are compared: level A includes original images resized with padding (Fig 3.A); level C adds data augmentation to level A (Fi. 3C); level D includes background removal, horizontal alignment, cropping and resizing without padding, plus augmentation (Fig 3D); and level G applies the complete preprocessing pipeline (Fig 3G). Panels 5A and 5B show results from the novel-device experiment, in which models were trained exclusively on images captured with one device type, i.e., either smartphone (Panel 5A) or microscope (Panel 5B), and tested on images from both the same or the other, previously unseen device. Panels 5C and 5D show results from the bias experiment, where training data were biased by linking species to specific imaging devices. In this setting, models were tested both on matching species-device combinations and on previously unseen combinations, i.e., either performance test on smartphone (Panel 5C) or microscope images (Panel 5D).

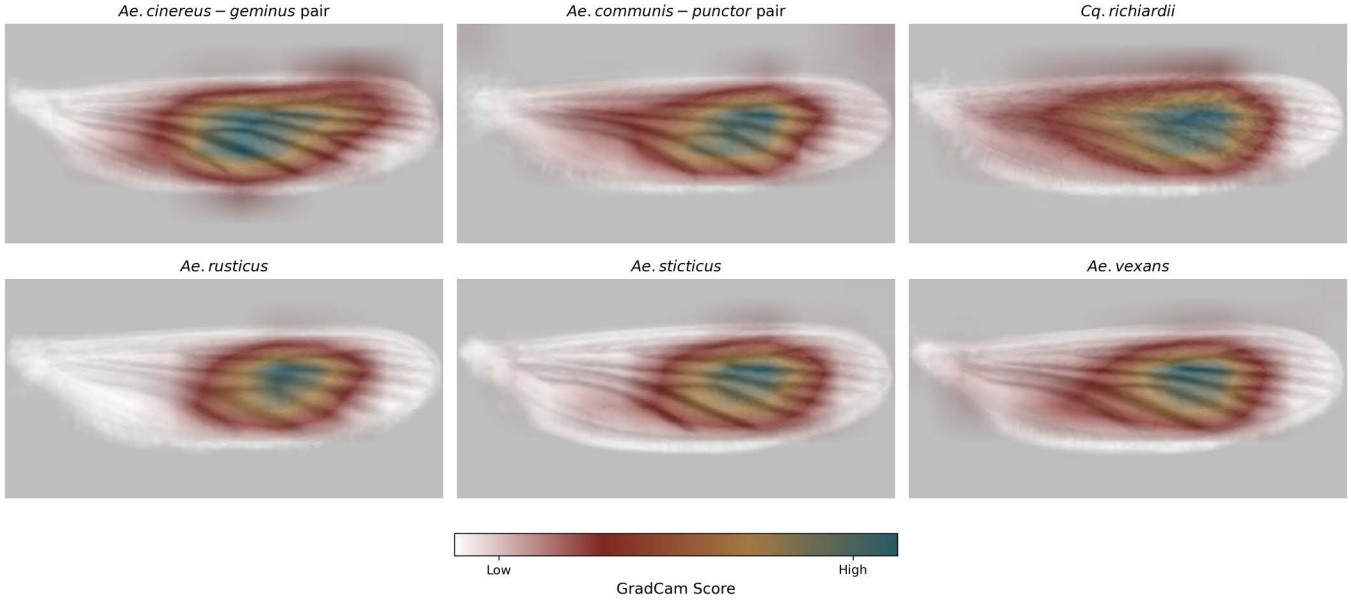

**Fig 6. Averaged GradCam heatmaps generated from 12 wing images for six mosquito taxa labels.** The heatmaps highlight the regions of the wings that the CNN model focuses on for classification, with blue areas indicating higher importance and yellow to red areas indicating lower importance while white areas are of no importance.

and edges) are less prominently highlighted. Notably, species such as *Cq. richiardii* and *Ae. cinereus-geminus*-pair exhibit broader and more diffuse activation patterns, compared to the tighter focal regions observed for taxa like *Ae. rusticus* and *Ae. sticticus* (Fig 6).

To explore the distribution of taxa within the feature space, UMAP analysis was performed on the feature maps of the testing fold 2 for the corresponding CNN model. The results reveal distinct clustering patterns, with most taxa forming clearly defined and well-separated clusters (Fig 7). Instances of proximity between clusters were observed, particularly among morphologically similar taxa, e.g., the close positioning of *Ae. aegypti* and *Ae. albopictus* or *Ae. koreicus* and *Ae. japonicus*. These observations align with the morphological challenges inherent in distinguishing these taxa, as their wing features exhibit significant similarity. Additionally, images belonging to the "other" species label form a broad, less cohesive cluster, reflecting the heterogeneous nature of this category. The UMAP analysis can also illustrate the influence of imaging devices on feature space organization (Fig 7) where no distinct clusters corresponding to different capture devices are observed.

### 3.4 Feasibility study

The feasibility study aimed to evaluate the applicability of the developed mosquito classification system in a real-world scenario. During the study, four participants were tasked to apply the developed models for 86 samples of mosquito wings which were neither represented in the training nor testing datasets. Hereafter all wings were imaged on a device not represented in the data. Example images from the two devices can be inspected in Fig A in S1 File. On the device represented in the dataset the five models demonstrated an average balanced accuracy of 96.4% (CI95%: 95.9 - 97.0) (Fig 8). The average balanced accuracy is 1.4% lower than the expected 97.8% based on the average taxa label accuracy on the testing set. On the novel device the models demonstrated an average balanced accuracy of 95.9% (95% CI: 95.0 - 96.9), which is comparable to both the expected and the performance on the known device. For the known device, accuracy ranged from 94.1% (worst) to 97.7% (best) across participant-model combinations. For the novel device, it

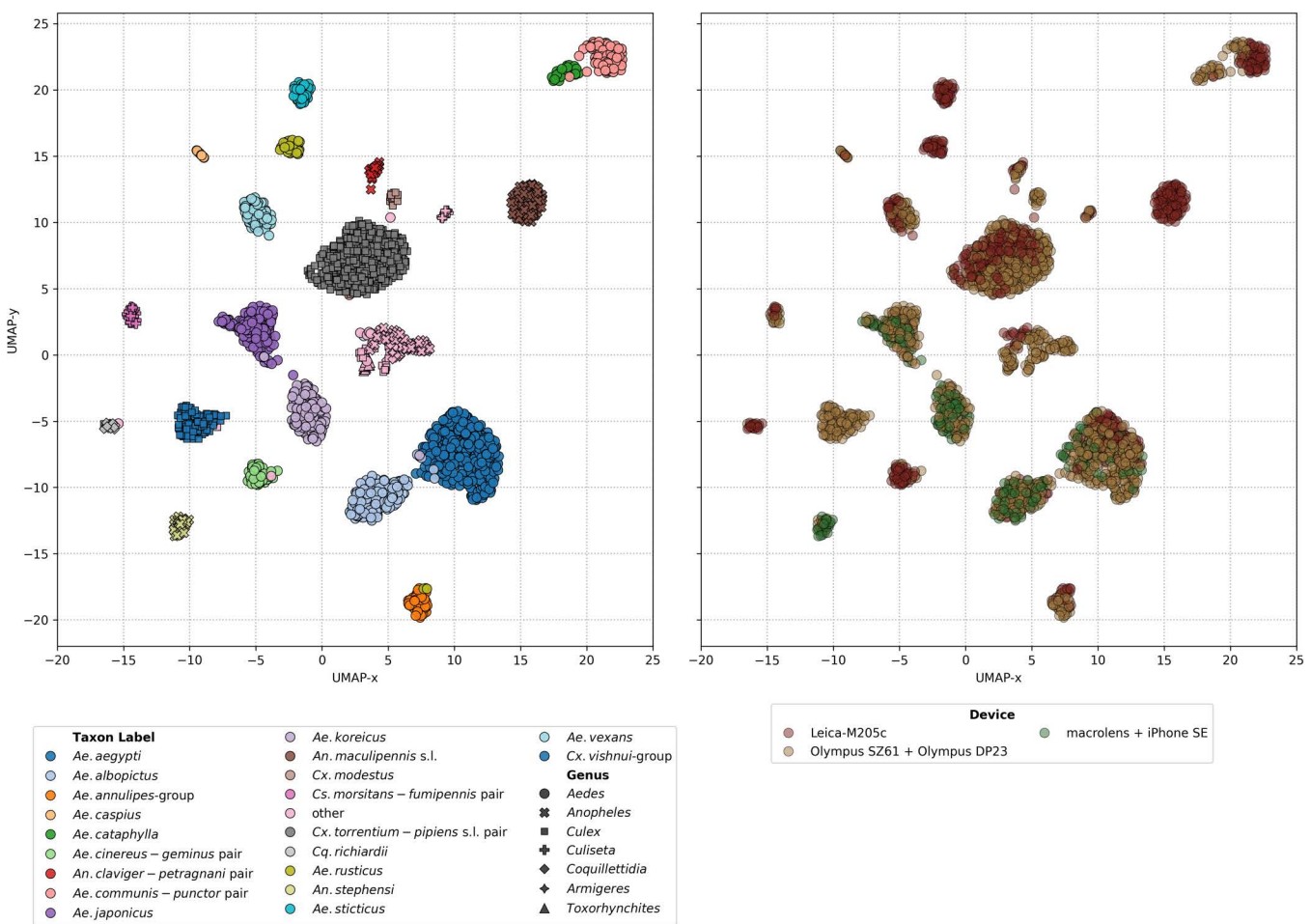

**Fig 7. UMAP on the feature maps of the training fold 2 for the corresponding CNN model.** Mosquito taxa colour-coded (left) and imaging devices-coloured (right).

ranged from 94.3% to 96.6%. Using the known and the unknown imaging devices, the worst performance was observed for *Cq. richiardii* which was only classified correctly with 72.5% and 62.0% accuracy, respectively. Most misclassifications were concentrated in two samples, which were consistently misclassified over all participants and models. There was no apparent reason for the complete misclassification of these samples. Despite these errors, the model achieved consistent predictions regardless of the participant and device. The Fleiss' Kappa coefficient, calculated for all predictions on the known device divided by participants, was 0.97, indicating a very high level of agreement of the predictions.

## 4 Discussion

This study investigated the application of CNNs to identify mosquito species based on wing images for vector surveillance and research. The objective of the study was not only to develop a reliable CNN model for the classification of mosquito wings but also to ensure that the model was applicable under real-world settings, including compatibility with different imaging devices and users. The developed classification models demonstrated a high performance both, measured in average balanced accuracy (98.2%) and average macro F1-score (97.6%). Notably, the models showed high performance across 21 taxa labels even with morphologically similar pairs, e.g., *Ae. aegypti* and *Ae. albopictus* or *Ae. koreicus* and *Ae.*

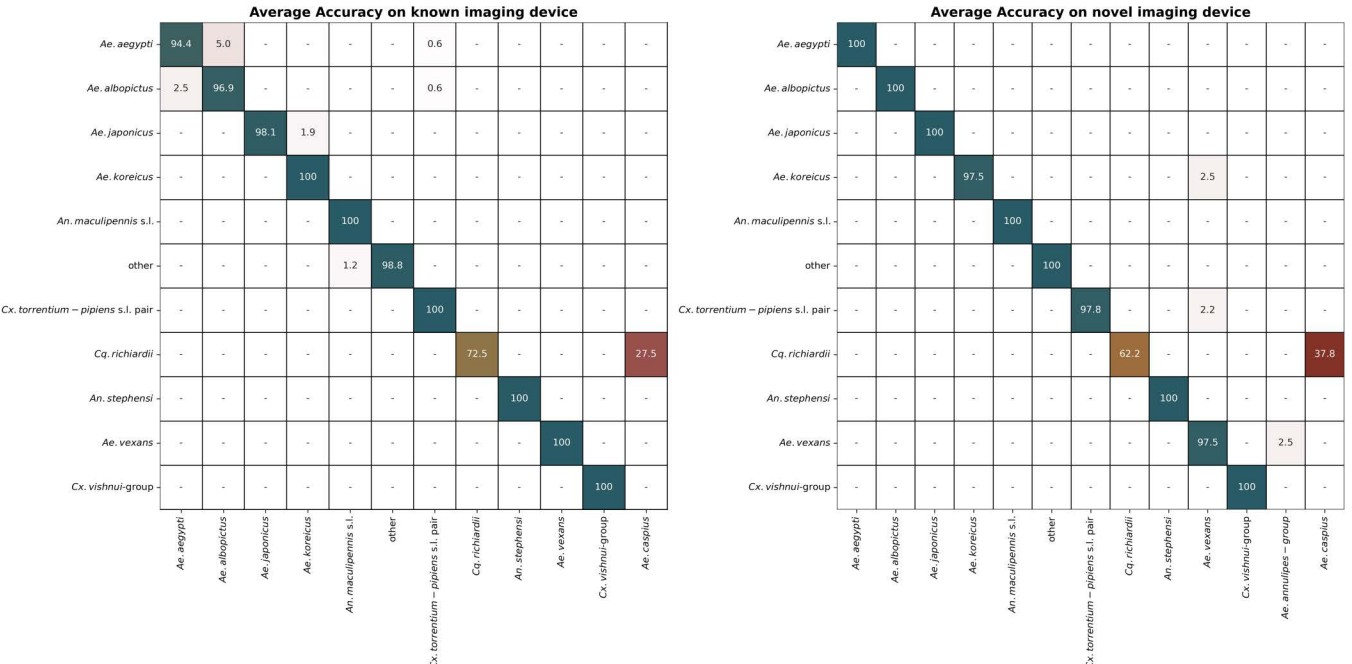

**Fig 8. Confusion matrix presenting the results of the feasibility study.** On the left performance from 4 different participants and 5 models were averaged. Images were taken on a device for which the models were trained for. On the right performance for 5 models were averaged. Images were taken on an imaging device which was not represented in the training set.

*japonicus*. The GradCam images indicate that the models are using reasonable features for wing classification. Additionally, the UMAP visualisation of a feature map shows that the model is grouping the different species labels into distinct clusters, indicating that the model has learned similar features from the images of the same species label. However, since UMAP is sensitive to initialization and parameter settings, its visual clusters should be interpreted with caution [34]. The most frequently misclassified species labels were *Cq. richiardii*, which had one of the fewest images in the dataset (N = 104), and the 'other' label, which included a diverse composition of less frequently represented taxa.

Compared to previous studies using neural networks for wing-based mosquito identification, our models demonstrated better performance both in the number of taxa classified and accuracy relative to the findings reported in the existing literature [19,35,36]. Even when compared to species identification through morphology the classification presents satisfying performance, e.g., Rahola et al. (2022) reported an average accuracy of just 81.5% for species identification based on responses from 51 participants provided with detailed images of mosquito [37] samples. This difference highlights the potential of CNN-based identification methods to surpass the limitations of traditional morphology-based approaches, which heavily depend on the expertise of the observer. However, it is important to note that our model's performance is inherently restricted to the species it was trained on, and its utility in broader applications will depend on the availability of training data for additional species.

The image preprocessing pipeline proposed in this study enhances the model's applicability to new devices that were not included in the training data. In our novel-device experiment, we found that increasing levels of preprocessing improved accuracy on new capture devices from 23-34% without preprocessing to 90–93% with full preprocessing, which is comparable to the model's accuracy on images from devices included in the training data. At the same time, preprocessing does not compromise the model's accuracy on the images that were captured with familiar devices, indicating that no essential features for classification are lost during preprocessing. While data augmentation is often recommended

to improve the generalizability of models to new settings [38,39], our experiments revealed that it only partially enhances adaptability to new devices and does not improve robustness to bias. This aligns with the hypothesis of Azulay and Weiss (2019) which argue that further augmentation during training does not lead to increased generalizability but only teaches the model to be invariant to the augmentations on the images visually similar to the ones from the training set. Moreover, the argument for preprocessing is further supported by Fujisawa et al. (2023), who identified the lack of standardization as a key factor contributing to reduced accuracy when applying CNN models beyond their original training distributions.

However, the bias reduction experiment still highlights some limitations of the preprocessing pipeline as it could only partially mitigate the effects of the highly skewed training dataset used in the experiments with a strongly biased distribution between two very different image capture methods. These models only achieved an average balanced accuracy of 35.8% on novel species-device combinations. Consequently, one must exercise caution when adding new species to the model's training data as device-specific variations in image characteristics may introduce biases that the current preprocessing pipeline cannot fully mitigate. We recommend adhering to a threshold of at least 100 images per species to ensure reliable training, consistent with our previous findings and our observations for *Cq. richiardii* [26]. For rare or underrepresented species, grouping them under the "other" label offers a practical alternative. This strategy would allow the model to reject the predictions, thereby reducing false positives while preserving overall reliability. In future research, the issue of a device-specific bias could be addressed by out-of distribution detection methods [40] or utilising contrastive learning, which trains the model to learn an embedding space where similar data points, i.e., species labels, are close together and dissimilar points are far apart [41]. Alternatively, modern style transfer techniques have been shown to be an effective approach for mitigating dataset bias [42].

Wings, as opposed to bodies, appear to be more reliable for this classification task. We applied heavy preprocessing methods, which most likely would have destroyed important features for automatic full mosquito body classification, e.g., omitting legs during background removal. While the removal of wings is an additional work step, wing vein patterns themself present as robust features, which are not associated with colour or texture, allowing to reduce the presence of undesirable features in the images without reducing the model performance. Furthermore, the depiction of whole mosquitoes can vary significantly due to factors such as age, physiological status (e.g., blood-fed or gravid), storage conditions, and image angle introducing additional variability that complicates the classification process [12,43]. Thus, the relatively stable and distinct structures of the wings probably provide a more reliable basis for preprocessing and subsequent classification.

While the presented classification model demonstrates strong performance, it is currently restricted to a limited selection of mosquito taxa primarily native to Europe [44]. This limitation introduces uncertainty regarding the model's ability to identify previously unencountered species. Specifically, the "other" label, designed as a catch-all category, is unlikely to adequately detect entirely new species due to the relatively narrow representation of taxa within the "other" label distribution [45]. Additionally, the wing shape and size within a species can slightly vary across mosquito populations from different regions, influenced by environmental and genetic factors [46–48]. Similarly, Fujisawa et al. (2023) demonstrated that CNNs trained on globally distributed images of beetles (*Coleoptera*) exhibit reduced performance when applied to a new, local dataset. While the extent to which this effect applies to mosquito wings is unknown, we anticipate that the models presented here may also demonstrate a performance decline when tested on unfamiliar local populations. Together, these factors underscore the importance of validating the classification system under novel conditions and, when necessary, tailoring it to specific ecological contexts.

The feasibility study supports the classification system's usability under real-world conditions, demonstrating high inter-rater reliability and a similar balanced accuracy compared to the testing performance. The ability of the preprocessing pipeline to enhance the model's adaptability to previously unencountered devices and different users highlights its potential for application in vector surveillance and research. The preprocessing methods reduce the need for stringent standardization protocols or specific image devices and address the needs of current vector surveillance and research, targeting

both major hurdles: lack of funding and trained personnel [5,6]. The here presented method only requires tweezers, and a macro-lens attached to a smartphone and can be conducted by a non-expert user. This effectively reduces the barriers to integrating this method into vector surveillance efforts and could facilitate its use in resource-constrained settings [49]. Additionally, this method will allow for the rapid identification of mosquito species and therefore has the potential to improve the efficiency of vector surveillance, enable timely public health interventions, and enhance disease outbreak preparedness in endemic regions. In the future, the model for mosquito wing classification can be extended to other dipteran insects of interest, e.g., biting midges or sandflies. These insects also exhibit distinct wing patterns that can be captured and analysed using similar preprocessing and classification techniques [7,50].

## Supporting information

**S1 File. Supporting Information. A file that contains all supporting information for the paper.** Table A. Glossary. Table B. Hyperparameters. Table C. Taxa Labels and Taxonomic IDs. Table D - F. Robustness experiment data distribution. Table G. Metrics. Fig A. Examples images from the Feasibility Study.
(DOCX)

## Acknowledgments

The authors want to acknowledge Juliane Bönecke for her insightful ideas, and valuable discussions, and Jonathan Ströbele for his support in realizing the application.

## Author contributions

**Conceptualization:** Kristopher Nolte, Christian Lins, Philip Kollmannsberger, Felix Gregor Sauer, Renke Lühken.

**Data curation:** Kristopher Nolte, Felix Gregor Sauer, Renke Lühken.

**Formal analysis:** Kristopher Nolte, Jens Johann Georg Lohmann, Felix Gregor Sauer.

**Funding acquisition:** Felix Gregor Sauer, Renke Lühken.

**Investigation:** Kristopher Nolte, Felix Gregor Sauer, Renke Lühken.

**Methodology:** Kristopher Nolte, Christian Lins, Philip Kollmannsberger, Felix Gregor Sauer, Renke Lühken.

**Project administration:** Renke Lühken.

**Resources:** Felix Gregor Sauer, Renke Lühken.

**Software:** Kristopher Nolte, Jens Johann Georg Lohmann, Philip Kollmannsberger.

**Supervision:** Jan Baumbach, Christian Lins, Philip Kollmannsberger, Felix Gregor Sauer, Renke Lühken.

**Validation:** Kristopher Nolte, Jan Baumbach, Christian Lins, Felix Gregor Sauer, Renke Lühken.

**Visualization:** Kristopher Nolte, Jens Johann Georg Lohmann.

**Writing – original draft:** Kristopher Nolte, Felix Gregor Sauer, Renke Lühken.

**Writing – review & editing:** Kristopher Nolte, Jan Baumbach, Christian Lins, Jens Johann Georg Lohmann, Philip Kollmannsberger, Felix Gregor Sauer, Renke Lühken.

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
