## [Decision Letter · Decision Letter 0]

17 Apr 2025

PCOMPBIOL-D-25-00450

Potentials and limitations in the application of Convolutional Neural Networks for mosquito species identification using wing images

PLOS Computational Biology

Dear Dr. Nolte,

Thank you for submitting your manuscript to PLOS Computational Biology. After careful consideration, we feel that it has merit but does not fully meet PLOS Computational Biology's publication criteria as it currently stands. Therefore, we invite you to submit a revised version of the manuscript that addresses the points raised during the review process.

Please submit your revised manuscript within 60 days Jun 17 2025 11:59PM. If you will need more time than this to complete your revisions, please reply to this message or contact the journal office at ploscompbiol@plos.org. Please include the following items when submitting your revised manuscript:

We look forward to receiving your revised manuscript.

Kind regards,

Fei Guo

Academic Editor

PLOS Computational Biology

Zhaolei Zhang

Section Editor

PLOS Computational Biology

**Journal Requirements:**

1)  Please upload all main figures as separate Figure files in .tif or .eps format. For more information about how to convert and format your figure files please see our guidelines: 

2) Please confirm whether your study includes live participants. If so, please insert an Ethics Statement at the beginning of your Methods section, under a subheading 'Ethics Statement'. It must include:

i) The full name(s) of the Institutional Review Board(s) or Ethics Committee(s)

ii) The approval number(s), or a statement that approval was granted by the named board(s)

iii) A statement that formal consent was obtained (must state whether verbal/written) OR the reason consent was not obtained (e.g. anonymity). 

3) Some material included in your submission may be copyrighted. According to PLOSu2019s copyright policy, authors who use figures or other material (e.g., graphics, clipart, maps) from another author or copyright holder must demonstrate or obtain permission to publish this material under the Creative Commons Attribution 4.0 International (CC BY 4.0) License used by PLOS journals. Please closely review the details of PLOSu2019s copyright requirements here: PLOS Licenses and Copyright. If you need to request permissions from a copyright holder, you may use PLOS's Copyright Content Permission form.

Potential Copyright Issues:

i) Please confirm (a) that you are the photographer of 3, or (b) provide written permission from the photographer to publish the photo(s) under our CC BY 4.0 license.

4) Please amend your detailed Financial Disclosure statement. This is published with the article. It must therefore be completed in full sentences and contain the exact wording you wish to be published.

5) Please provide a completed 'Competing Interests' statement, including any COIs declared by your co-authors. If you have no competing interests to declare, please state "The authors have declared that no competing interests exist". 

**Reviewers' comments:**

Reviewer's Responses to Questions

Reviewer #1: The authors present a manuscript detailing the application of convolutional neural networks (CNNs) for identifying mosquito species using wing images. The introduction effectively highlights the need for a reliable and user-friendly system that can be deployed in resource-limited regions. The authors provide a comprehensive description of the preprocessing of wing images, the training of their CNNs (based on the EfficientNet architecture), and the results of testing these models. The reported accuracy is impressively high, rarely dropping below 97% (Figure 4). Additionally, the authors commendably make their code freely available, although I encountered some difficulties in executing it (see below).

However, I have several concerns regarding the manuscript in its current form. Firstly, the high accuracy of the models suggests potential overfitting to the training data. The authors acknowledge this issue in Figure 5, noting that models performing well on familiar data may struggle with new data. To demonstrate the models' practical utility, Section 4.4, which describes a feasibility study, needs substantial expansion. Furthermore, the feasibility study used the same imaging system (Olympus SZ61) as most of the training data; it would be beneficial to explore how the models perform with different imaging systems.

My most significant concern is the manuscript's similarity to two previously published works by a similar group of authors from last year:

* https://parasitesandvectors.biomedcentral.com/articles/10.1186/s13071-024-06459-3

* https://www.nature.com/articles/s41598-024-53631-x.

Both papers describe similar approaches using CNNs to identify mosquito species based on wing images, with one even utilizing the same EfficientNet architecture. The authors should acknowledge these previous works more directly in their manuscript (they are currently only cited as the source of their images) and clarify how the current study represents a significant advancement. The primary difference appears to be the inclusion of a larger number of mosquito species. The introduction should be revised to clearly state that this work builds upon previously published research by the same authors. Additionally, the introduction lacks references to the shortcomings of other published works. Overall, a much stronger case needs to be made for the novelty and significance of the current study.

Other points:

* Discussion of the labelling strategy for training data in the methods is a little confusing and Figure 1 is not very informative. I think a table summarising the exact composition of all data used for training, validation and testing would be helpful

* The distinction between what constitutes preprocessing and augmentation could be clearer. Augmentation is for training purposes only, but preprocessing is always necessary. For example, this statement (lines 162-163) is confusing: "During training the augmentation is used to increase the variance of the training set, during deployment the augmentation is used for test-time augmentation, only during testing this step is omitted"

* The Pre-processing seems quite involved - I would be surprised if it works reliably with unseen data. Have the authors tried training the classifier without the preprocessing step? Because this might generalise better?

* No real explanation is offered as to why EfficientNet was used? Did the authors try any other architectures? Has anyone else previously published other models used for similar purposes?

* Essentially, 5 different models were trained and the results presented show the average result returned by the 5 models. But how much variability was there across the five models?

* Lines 210 - 214 are confusing. Again, a table listing the details of the training data would be more useful here.

* Lines 215 - 222: I have to be honest, I'm getting a little lost here - I have no idea (a) what the aim is here or (b) what is being used for training data and what is being used for testing. We seem to have gone off on a tangent.

* In the "Robustness experiments" section, is the training data referred to here completely distinct from that referenced earlier? Or is it a subset? I ask, because the description of the dataset seems to closely match characteristics shown in Fig 1.

* There are multiple references throughout the paper to different training sets - it is not totally clear what training data was actually used to generate each of the presented results. I would suggest having a table describing exactly what data was used for training for each Figure.

* It's great that the authors have made code openly-available, but there is little in the way of installation assistance in the Github repo. I tried setting up the necessary python environment to run the notebooks provided (https://github.com/KNolte19/MosquitoWingClassifier_publication), but even after correcting some apparent typos in the requirements file, the pip installation still failed:

ERROR: Could not find a version that satisfies the requirement tensorflow-macos==2.14.0 (from versions: none)

ERROR: No matching distribution found for tensorflow-macos==2.14.0

Reviewer #2: In this manuscript, the authors developed methods to identify mosquito species from wing images by a CNN model trained with a large curated database. They tested the robustness of identification under different imaging devices and built a data augmentation pipeline to improve model's generalization capability. Authors also tested model's performance in a feasibility study.

The data augmentation successfully reduced the identification errors introduced by device differences, but when the strong correlation between species and devices existed, generalization was difficult.

While many accurate models for mosquito / insect identification by deep learning exist, they do not often consider the broad generalization capability, which is critically important when these methods are rolled out for real public surveillance. This study successfully mitigate the performance drop caused by the difference of capturing devices and showed some conditions where generalization is extremely difficult. I think these results help researchers design effective strategies for machine learning-based vector surveillance.

The experiments are carefully designed to show the advantages and limitations of their method. The model architecture is standard and I do not find critical flaws in methodological aspects. The presentation of the manuscript is good. All parts are concisely presented and results are sufficiently discussed. Therefore, I think this manuscript in good quality, and I only have minor points for revision.

My only concern is the scope of the manuscript. I feel it is slightly more application-oriented than standard PLOS Computational Biology articles.

Minor points:

Line 117: Please report information of the capturing devices, for example, proportions of different devices.

Line 338: Adding a UMAP visualization of trials without applying the pipeline may help show the effectiveness of the current methods. I guess that the clusters will separated based both on species and devices if the pipeline is not applied.

Line 416 image "angle"?

Line 423: Application of out-of-distribution detection methods may be a future direction.

Yang et al. 2024 https://link.springer.com/article/10.1007/s11263-024-02117-4

It may improve model performance since a large portion of errors are due to errors over the "other" category.

Figure 5: "Level D" is missing in the caption of this figure (line 303). Also, captions are a bit confusing. Does "level D" include data augmentation or does it apply only image processing without augmentation?

**Have the authors made all data and (if applicable) computational code underlying the findings in their manuscript fully available?**

Reviewer #1: Yes

Reviewer #2: Yes

PLOS authors have the option to publish the peer review history of their article (what does this mean? ). If published, this will include your full peer review and any attached files.

**Figure resubmission:**
---

## [Decision Letter · Decision Letter 1]

9 Jul 2025

PCOMPBIOL-D-25-00450R1

Potentials and limitations in the application of Convolutional Neural Networks for mosquito species identification using wing images

PLOS Computational Biology

Dear Dr. Nolte,

Thank you for submitting your manuscript to PLOS Computational Biology. After careful consideration, we feel that it has merit but does not fully meet PLOS Computational Biology's publication criteria as it currently stands. Therefore, we invite you to submit a revised version of the manuscript that addresses the points raised during the review process.

Please submit your revised manuscript within 60 days Sep 08 2025 11:59PM. If you will need more time than this to complete your revisions, please reply to this message or contact the journal office at ploscompbiol@plos.org. Please include the following items when submitting your revised manuscript:

We look forward to receiving your revised manuscript.

Kind regards,

Zhaolei Zhang

Section Editor

PLOS Computational Biology

Zhaolei Zhang

Section Editor

PLOS Computational Biology

**Journal Requirements:**

1) Figure 3: Please confirm (a) that you are the photographer; or (b) provide written permission from the photographer to publish the photo(s) under our CC BY 4.0 license. 

**Reviewers' comments:**

Reviewer's Responses to Questions

**Comments to the Authors:**

Reviewer #1: I thank the authors for their point-by-point response to my previous comments. However, I’m afraid the revisions do not fully address the issues raised. The first two points below are particularly important, given the authors’ stated aim in the introduction to advance their previous work toward “practical, real-world deployment”:

1. Code Usability:

I appreciate the effort to update the instructions on the GitHub repository. However, it remains unclear how to run the code on a novel test image. For instance, if I capture an image of a mosquito wing using my phone, how do I test the classifier on this image? The README lists several Jupyter notebooks, but these lack detailed documentation. The test_classifier.ipynb notebook appears most relevant, yet it seems to require .npy files that have already been preprocessed. Clear, step-by-step instructions for running the model on raw images would be essential for real-world usability.

2. Feasibility Study and Benchmarking:

Section 4.4 remains limited and lacks accompanying figures. For example, including confusion matrices—one for a “known” imaging device and one for a “novel” device—would greatly enhance clarity. Additionally, there is no comparison with previously published models. Since the authors claim that "Traditional ML pipelines, while effective in many scenarios, frequently fall short in real-world deployment", it would be valuable to see how their model performs relative to the current state-of-the-art.

3. Table Listing Data Used:

It is unclear where the supplementary information referenced by the authors can be found. I previously asked to see a breakdown of how exactly the data was used for training, testing, validation, etc. in the form of a table. The authors have advised that I should refer to the Git repository, but I'm not sure what exactly I should be referring to. I have also been referred to Supplementary Table S4, but I could not locate any supplemental information. What I would like to see is something similar to Tables 1 and 2 here, for example: https://www.life-science-alliance.org/content/7/1/e202302351/tab-figures-data.

Reviewer #2: This is a revised version of the manuscript I reviewed. It reports a preprocessing pipeline for mosquito identification based on wing images.

The authors responded most of the points I raised in the previous review round. Also, they made substantial revisions on the texts and added a new analysis. In the last review, I found the manuscript was already in good quality. The current version is better in quality with added explanations of backgrounds and a new feasibility study, which shows the generalization capability of the model.

I only have very minor points to mention over the current version.

Minor point:

Line 450 (Line 416 in the previous version): "image ANGLE" is spelled as "image ANGEL".

GitHub repository:

Just like the reviewer 1, I encountered several errors before the codes properly ran. I think better documentations can lead to wider acceptance of this method by general users.

**Have the authors made all data and (if applicable) computational code underlying the findings in their manuscript fully available?**

Reviewer #1: Yes

Reviewer #2: Yes

PLOS authors have the option to publish the peer review history of their article (what does this mean? ). If published, this will include your full peer review and any attached files.

**Do you want your identity to be public for this peer review?** For information about this choice, including consent withdrawal, please see our Privacy Policy .

Reviewer #1: **Yes: ** David J Barry

Reviewer #2: No

**Figure resubmission:**
---

## [Decision Letter · Decision Letter 2]

13 Aug 2025

Dear Mr. Nolte,

We are pleased to inform you that your manuscript 'Potentials and limitations in the application of Convolutional Neural Networks for mosquito species identification using wing images' has been provisionally accepted for publication in PLOS Computational Biology.

Best regards,

Fei Guo

Academic Editor

PLOS Computational Biology

Zhaolei Zhang

Section Editor

PLOS Computational Biology

Reviewer's Responses to Questions

**Comments to the Authors:**

Reviewer #1: I thank the authors once again for their detailed responses to the points I raised. I have successfully run their new notebook (run_classifier_DEMO.ipynb) without issues and thank them for making this addition. I have even managed to test it on images downloaded from wingbank.butantan.gov.br and elsewhere and the performance is impressive.

I see no further issues, I commend the authors on their rigorous methodology and hope that their model finds widespread use in the field.

Reviewer #2: This is a second revised version of the manuscript I reviewed before. As I already said in the first and second review rounds, the manuscript is in good condition, and the current version of the repository contains enough information and data to fully reproduce the study. So, I do not have any further objections over this manuscript.

Only one point I would add is that it may be helpful if there is some information of the performance requirements in the code document. The procedures for training and analyses took quite long time on my laptop (Ryzen5 3Ghz 6cores + 16GB memory).

**Have the authors made all data and (if applicable) computational code underlying the findings in their manuscript fully available?**

Reviewer #1: Yes

Reviewer #2: Yes

PLOS authors have the option to publish the peer review history of their article (what does this mean? ). If published, this will include your full peer review and any attached files.

**Do you want your identity to be public for this peer review?** For information about this choice, including consent withdrawal, please see our Privacy Policy .

Reviewer #1: **Yes: ** David J Barry

Reviewer #2: No

---

## [Editor Report · Acceptance letter]

PCOMPBIOL-D-25-00450R2

Potentials and limitations in the application of Convolutional Neural Networks for mosquito species identification using wing images

Dear Dr Nolte,

I am pleased to inform you that your manuscript has been formally accepted for publication in PLOS Computational Biology. Your manuscript is now with our production department and you will be notified of the publication date in due course.

With kind regards,

Benedek Toth
